# The Acute and Chronic Cognitive Effects of a Sage Extract: A Randomized, Placebo Controlled Study in Healthy Humans

**DOI:** 10.3390/nu13010218

**Published:** 2021-01-14

**Authors:** Emma L. Wightman, Philippa A. Jackson, Bethany Spittlehouse, Thomas Heffernan, Damien Guillemet, David O. Kennedy

**Affiliations:** 1Brain Performance and Nutrition Research Centre, Northumbria University, Newcastle Upon Tyne NE1 8ST, UK; philippa.jackson@northumbria.ac.uk (P.A.J.); b.spittlehouse@northumbria.ac.uk (B.S.); tom.heffernan@northumbria.ac.uk (T.H.); david.kennedy@northumbria.ac.uk (D.O.K.); 2NUTRAN, Northumbria University, Newcastle Upon Tyne NE1 8ST, UK; 3Nexira SAS, 129 Chemin de Croisset-CS 94151, 76723 Rouen, France; d.guillemet@nexira.com

**Keywords:** sage, *Salvia officinalis*, *Salvia lavandulaefolia*, polyphenols, cognition

## Abstract

The sage (*Salvia*) plant contains a host of terpenes and phenolics which interact with mechanisms pertinent to brain function and improve aspects of cognitive performance. However, previous studies in humans have looked at these phytochemicals in isolation and following acute consumption only. A preclinical in vivo study in rodents, however, has demonstrated improved cognitive outcomes following 2-week consumption of Cognivia^TM^, a proprietary extract of both *Salvia officinalis* polyphenols and *Salvia lavandulaefolia* terpenoids, suggesting that a combination of phytochemicals from sage might be more efficacious over a longer period of time. The current study investigated the impact of this sage combination on cognitive functions in humans with acute and chronic outcomes. Participants (*n* = 94, 25 M, 69 F, 30–60 years old) took part in this randomised, double-blind, placebo-controlled, parallel groups design where a comprehensive array of cognitions were assessed 120- and 240-min post-dose acutely and following 29-day supplementation with either 600 mg of the sage combination or placebo. A consistent, significant benefit of the sage combination was observed throughout working memory and accuracy task outcome measures (specifically on the Corsi Blocks, Numeric Working Memory, and Name to Face Recall tasks) both acutely (i.e., changes within day 1 and day 29) and chronically (i.e., changes between day 1 to day 29). These results fall slightly outside of those reported previously with single *Salvia* administration, and therefore, a follow-up study with the single and combined extracts is required to confirm how these effects differ within the same cohort.

## 1. Introduction

The Nepetoideae subfamily of the Lamiaceae family of plants, which provides most of our culinary herbs and many essential oils, is a particularly rich source of plants that use volatile terpenes in symbiotic ecological roles. Members include psychoactive herbs such as rosemary, lemon-balm, peppermint and sage, and this group typically synthesise mono- and sesquiterpenes such as 1,8-cineole, α-pinene, camphor, geraniol, geranial, borneol, camphene and β-caryophyllene. These plants also express high levels of phenolics including, in all cases, rosmarinic acid and its derivatives, alongside other phenolic acids and polyphenols [1].

Extracts from this family of plants often share common (but variable) mechanisms of action relevant to the brain, for instance, inhibition of acetylcholinesterase and binding allosterically to gamma-Aminobutyric acid (GABA^A^), nicotinic and muscarinic receptors [1]. These effects are most likely attributable to the terpene content outlined above [2,3,4,5,6]. The phenolic constituents have also demonstrated effects pertinent to brain function, e.g., increases in brain-derived neurotrophic factor (BDNF), and antioxidant and anti-inflammatory activities in neurons (see [7] for a review). More importantly, a recent trial by our own lab demonstrated the more robust effects (in this case, acutely on energy and mood in humans) which can be achieved when combining phenolics (coffee berry, apple catechins and blueberry anthocyanins) and terpenes (sage and ginseng) into a single supplement [8].

The psychoactive effects of consuming single doses of extracts and dried leaf of *Salvia officinalis* and *Salvia lavandulaefolia* (common and Spanish sage, respectively), containing the above-defined full spectrum of phytochemicals, also extend to cognitive function. (It is interesting to note here that, while there are some similarities in the phytochemical composition of sage species, differences also exist. For instance, *salvia officinalis* and *lavandulaefolia* can contain similar percentages of 1,8-cineole but significantly different levels of camphor (22.2% versus 19.6% 1,8-cineole, respectively, and 33.6% versus 15.5% camphor, respectively) [3,9]. With regards to phenolic composition, the two species seem broadly in line although *salvia officinalis* appears to synthesise salvianolic acid derivatives of rosmarinic acid where *salvia lavandulaefolia* does not [1].) Here, vigilance and memory in both younger and older adults seem to be most sensitive to improvements [4,5]. Several studies have also assessed the effects of a single dose of essential oils composed solely of the volatile terpenes present in *Salvia lavandulaefolia* plant material. In the first of these studies, healthy young participants took single oral doses of 50 and 100 µL of encapsulated *salvia officinalis* dried leaf [4] and exhibited improvements in memory (immediate and delayed recall of words) within the first 2.5 h post-dose. Similar mnemonic effects were subsequently confirmed following single doses of 25 and 50 µL of the same essential oil along with improved performance on a mental arithmetic task and improved levels of subjective alertness, calmness and contentment [10]. Most recently, the psychoactive properties of *Salvia lavandulaefolia* were confirmed in healthy young adults who consumed single doses of essential oil that exclusively contained monoterpenes. Specifically, this oil provided a high concentration of 1,8-cineol and presented a particularly potent acetylcholinesterase inhibitory profile. Within the first four hours, single doses improved memory and attention task performance, increased alertness and reduced mental fatigue during extended performance of difficult tasks [11].

Taken together, these human randomised controlled trials demonstrate robust and consistent acute cognitive effects of *Salvia* when administered as a single species (either *officinalis* or *lavandulaefolia*). However, what is not known is whether chronic effects could also be elicited nor whether, when the species are co-administered, these effects pervade and/or synergistic mechanisms of action produce different effects entirely. This question was partly answered recently by a preclinical in vivo rodent study where a combination of essential oil from *Salvia lavandulaefolia* and a leaf extract from *Salvia officinalis* (Cogniva™) was investigated [12]. Here, the combination was compared to each species alone, and a control, which demonstrated a significant acute effect on memory in the Y-maze after all single doses of *Salvia* and a significant chronic (following 2 weeks of treatment) effect on visuospatial memory in the Morris water maze following the combination only. Biochemical and histological investigation after the end of the administration period revealed that sage stimulated the expression of calcium–calmodulin-dependent protein kinase II (CaMKII), a mechanism that has been proposed to regulate the biochemical neuronal processes supporting working memory, learning and interpretation [13,14,15,16].

Taken together, the original research question addressed by this study is whether this *Salvia* and phenolic combination can produce similarly positive effects on memory in humans and how it might mediate persistent attentional and memory demands across various media, in particular, in an attempt to mirror real-world demands.

## 2. Methods

### 2.1. Study Design and Participants

The study aimed to analyse a final data set of *n* = 90 (*n* = 45 per condition) healthy adults from this randomised (simple), double-blind, placebo-controlled, parallel groups design. A secondary aim was to balance the age of participants across 3 age categories: *n* = 30 between 30–40 years, *n* = 30 between 41–50 years and *n* = 30 between 51–60 years. This was, in large part, to help balance the age of participants across this broad range, and it was hoped that, in doing so, analyses could take into account potential age differences (However, as it transpired, no interpretable interactions between treatment × age were revealed in the analyses (likely because the analysis simply was not powered to include this), and therefore, it was decided that, in order to maintain clarity with so many other outcomes, the reported analyses omit this factor.) Participants were recruited from the local area via social media, internal email for Northumbria University staff and students, and advertisements in the local newspaper. The actual number recruited, to allow for any loss pre-analysis, was *n* = 94 (69 female and 25 male, mean age 43.9 years (SD 8.6 years), 88 right-handed and 6 left-handed, mean years in education 16.7 years (SD 3.6 years) and mean BMI 25.4 (SD 3.5)). The number of participants in the placebo condition was 49, and 45 participants consumed the active intervention. *n* = 36 were aged 30–40 years, *n* = 32 were between 41–50 years and *n* = 26 were between 51–60 years. See Figure 1 for participant disposition diagram and Table 1 for demographic breakdown for each treatment condition.

Participants were excluded if they had any pre-existing medical condition/illness which would impact taking part in the study (this was necessarily vague to encompass any unforeseen issues on a case-by-case basis); were currently taking prescription medications which would contraindicate with the study (i.e., which might impact the outcome measures directly themselves); had high blood pressure (systolic over 159 mm Hg or diastolic over 99 mm Hg); had a BMI outside of the range 18.5–30 kg/m^2^ (the exclusion of those in the underweight and obese BMI ranges was an attempt to mitigate against any potentially excluding health complaints associated with these extreme ends of adiposity); were pregnant, seeking to become pregnant or lactating; had learning and/or behavioural difficulties such as dyslexia or ADHD; had a visual impairment that could not be corrected with glasses or contact lenses (including colour-blindness); smoked; consumed excessive amounts of caffeine (>500 mg per day); had food intolerances/sensitivities; had taken antibiotics, prebiotics or probiotics (including drinks, e.g., Yakult^®^ or Actimel^®^) within the past 8 weeks; were currently participating in other clinical or nutrition intervention studies or had in the past 4 weeks; had been diagnosed with/undergoing treatment for alcohol or drug abuse in the last 12 months; had been diagnosed with/undergoing treatment for a psychiatric disorder in the last 12 months; suffered from frequent migraines that require medication (more than or equal to 1 per month); had sleep disturbances (including night-shift work) and/or were taking sleep aid medication; or had active infections.

This study was conducted according to the guidelines laid down in the Declaration of Helsinki 1975, and all procedures were approved by the department of Psychology (Northumbria University) staff ethics committee (code: 8970). Written informed consent was obtained from all participants.

### 2.2. Treatments

Participants consumed either placebo or 600 mg Cognivia^TM^ (a proprietorial supplement (with more detail available in [12]) which is available to buy over the counter) every day for 29 +/− 3 days (This was the permitted range for those who could not return on day 29, but in actuality, nobody was under 29 days). The selected dose contained 400 mg of aqueous extract from *Salvia officinalis* leaves characterized for its content in polyphenols (as rosmarinic acid, apigenin glucosides, luteolin glucosides and others). The remaining 200 mg contained 50 µL of *Salvia lavandulaefolia* essential oil characterized for its content in terpenoids (as eucalyptol, camphor, α- and β-pinene, and others) and encapsulated with gum acacia. The posology of both active substances was selected in accordance with descriptions of the most effective dosages described in the clinical acute studies introduced above (i.e., 50 µL of *Salvia lavandulaefolia* essential oil and extract of *Salvia officinalis* with a ratio equivalent to 2.25 g of dried leaves). The encapsulated powder of essential oils with gum acacia has been proposed to facilitate posology and observance compared to the liquid form and to protect terpenoids from evaporation and oxidation. Acacia gum has a long history of use with terpenes and essential oil protection [17], and the encapsulation has not compromised the nootropic activities of *Salvia lavandulaefolia* essential oil in a previous preclinical study [12]. Treatment was in the form of blue capsules (both active and placebo) and was dispensed from identical white bottles; participants took their first and last lab-based doses from these bottles and self-supplemented at home in the interim. Two compliance measures were used to determine adherence to the treatment regimen; the primary measure was a capsule count of returned treatment, and the second was reference to a treatment diary in which the participant noted the time of their treatment consumption each day.

### 2.3. Cognitive, Mood and Blood Pressure (BP) Assessment

Computerised Mental Performance Assessment System (COMPASS):

This testing system delivers a bespoke collection of tasks, with fully randomised parallel versions of each task delivered at each assessment for each individual. The battery has been in use within Northumbria University for over 10 years; is now commercially available for other research organisations (www.cognitivetesting.com); and is currently in use within a number of UK, US, New Zealand and Australian Universities, companies and research organisations. Each battery is entirely self-contained, e.g., the stimuli presented at the start (i.e., the pictures and words) are recalled at the end and any subsequent batteries present a novel selection of stimuli. All tasks are briefly summarised below, and full descriptions can be found elsewhere [18].

#### 2.3.1. Picture Presentation

Fifteen colour photographic images of everyday objects (such as a telephone, car or cup) were presented sequentially on screen for the participant to remember at the rate of 1 every 3 s, with a stimulus duration of 1 s.

#### 2.3.2. Face Presentation

Twelve passport-style photographic images of people, containing a first and last name underneath, were presented sequentially in a random order to participants. Stimulus duration was 3 s, with a 1-s interstimulus duration.

#### 2.3.3. Word Presentation

Fifteen words were presented sequentially on screen for participants to remember. Stimulus duration and interstimulus time were both 1 s.

#### 2.3.4. Immediate Word Recall

The participant was allowed 60 s to write down as many of the words that were just presented as possible. The task was scored for number correct and errors.

#### 2.3.5. Numeric Working Memory

Five digits (1–9) were presented sequentially for the participant to remember. This was followed by a series of 30 probe digits (15 targets and 15 distractors), and the participant indicated whether it had been in the original series by a simple “yes” or “no” key press. The task consisted of 3 separate trials. Accuracy and mean reaction time for correct responses were recorded.

#### 2.3.6. Choice Reaction Time

Participants responded with a left or right key press corresponding to the direction of the arrows appearing on screen. The randomly varying interstimulus interval was between 1 and 3 s for a total of 50 stimuli. Accuracy and mean reaction time for correct responses were recorded.

#### 2.3.7. Corsi Blocks Task

A set number of blocks, from a maximum of 9, changed colour from blue to red in a randomly generated sequence and, once finished, participants were instructed to repeat the sequence by clicking on the blocks using the mouse and cursor. The task was repeated five times at each level of difficulty, from 4 upwards, until the participant could no longer correctly recall the sequence. The task was scored for “span score”, calculated by averaging the level of the last three correctly completed trials.

#### 2.3.8. Cognitive Demand Battery (CDB)

The CDB comprises repeated (in this case, 3 repetitions) of the Serial 3s subtraction task (2 min), Serial 7s subtraction task (2 min) and Rapid Visual Information Processing (RVIP, 5 min) task. For the serial subtraction tasks, participants subtracted either 3 or 7 consecutively from a randomly generated number between 800 and 999 for the duration of the task, entering their responses on the keyboard’s linear number pad. These tasks were scored for the number of correct subtractions and the number of errors. For the RVIP task, participants monitored a continuous series of single digits (1–9 at a rate of 100 per minute) for targets of three consecutive odd or three consecutive even digits (8 per minute). The task was scored for number of correctly identified target strings and average reaction time for correct detections.

#### 2.3.9. Peg and Ball Task

Participants were presented with 2 configurations of 3 coloured balls (blue, green and red) on 3 pegs that each hold 3 balls. Participants had to rearrange the balls, moving one ball at a time, from the starting configuration so that they matched the position of the balls in the goal configuration. Each trial (of 5) generated scores for planning times prior to moving, time to completion and errors.

#### 2.3.10. Delayed Word Recall

The participant was again given 60 s to write down as many of the words presented previously as possible. Total number of correct responses and errors were recorded.

#### 2.3.11. Delayed Word Recognition

The original 15 words, plus 15 distractor words, were presented one at a time in a random order with participants responding “yes” or “no” as to whether they were originally presented. Accuracy and mean reaction time for correct responses were recorded.

#### 2.3.12. Delayed Picture Recognition

The original 15 pictures plus 15 distractor pictures were presented one at a time in a random order with participants responding “yes” or “no” as to whether they were originally presented. Accuracy and mean reaction time for correct responses were recorded.

#### 2.3.13. Name-to-Face Recall

The twelve original photographs presented at outset were presented on the screen, one at a time with a list of 4 different first names and 4 different last names underneath. Participants had to choose the first and last name that was originally presented with the photograph. The numbers of correct responses for first and last names were recorded and collapsed to give an overall score for this task.

#### 2.3.14. Global Cognitive Measures

Almost all of the above cognitive tasks produce multiple outcome measures, e.g., a measure of accuracy, speed and error in performing the individual task. This allows one to investigate performance further by combining those same measures across all appropriate tasks. From this, global cognitive measures can then be derived: accuracy of attention (comprised of accuracy in relation to the choice reaction time and rapid visual information processing tasks), speed of attention (comprised of the reaction time performance in relation to the choice reaction time and rapid visual information processing tasks), working memory (comprised of accuracy in relation to the numeric working memory task and span score on the Corsi Blocks tasks), speed of memory (comprised of reaction time performance in relation to the numeric working memory, picture recognition and word recognition tasks) and episodic memory (comprised of accuracy in relation to the immediate and delayed word recall tasks, name-to-face recall, and picture and word recognition).

Figure 2 outlines the individual tasks used, the order in which they were presented and the approximate timings as well as the primary cognitive domain of each task (left-hand-side of the diagram). The figure also depicts the global cognitive domains (right-hand-side) and where the data was sourced from the individual cognitive tasks to derive these global scores.

#### 2.3.15. Prospective Remembering Video Procedure (PRVP)

The PRVP task, a location-learning task, has proven sensitive to memory failings during impaired states, e.g., during binge drinking [19]. Less is known about its sensitivity to normal memory degradation over time and whether it can detect attenuation of this in response to nutritional supplementation. As such, its use here is somewhat exploratory. The task required participants to encode a list of 18 locations and their matched action within 60 s. These location–action pairs then unfolded during a 10-min video clip of a walk down a shopping street, e.g., “At Thorntons, buy a bag of toffees.”, during which participants must note down as many pairs as possible, along with how many pushchairs they saw during the clip (acting as a distractor and not analysed). The performance of participants for their first completion of this task (i.e., day 25) was scored as previously described [19], i.e., 1 point for each correctly remembered location–action (with a potential total of 18 points), as this is the original procedure for completing the task. However, we also wanted to measure whether memory degradation of these locations–actions was affected by treatment and so, on day 29, participants completed this task again but without seeing the list of locations–actions first. Here, participants were awarded 1 point for each location and action (i.e., with a potential total of 36 points) as it was anticipated that remembering would be significantly more challenging when relying on encoding from 4 days ago, and the original scoring method may lead to floor effects.

#### 2.3.16. Mood

To assess mood, the current study used both the Bond–Lader [20] mood scales (completed at the beginning of each task battery repetition) and the State Trait Anxiety Inventory (STAI) [21]. Both state and trait anxiety were measured during the screening/training visit to provide a baseline measure of mood. Subsequently, during each testing session, only state was assessed as trait mood should be stable across this relatively short period.

#### 2.3.17. Blood Pressure

Sitting blood pressure and heart rate readings were collected using a Boso Medicus Prestige blood pressure monitor with the subject’s arm supported at the level of the heart and with their feet flat on the floor. Readings were taken following completion of the baseline tasks and again following completion of the post-dose tasks.

### 2.4. Procedure

Testing took place between August 2018–April 2019 at Northumbria University, UK, within a suite of testing facilities with participants visually isolated from each other. Participants attended the laboratory on 4 separate occasions: an introductory visit between 1 and 14 days before the first day of treatment, two testing days (day 1 and day 29) and an interim visit (day 25).

The introductory visit to the laboratory was comprised of briefing on the requirements of the study, obtaining informed consent, health screening, completing the Caffeine Consumption Questionnaire (CCQ) and State-Trait Anxiety Inventory (STAI) trait subscale, training on the cognitive and mood measures, and collecting demographic data.

For the two ensuing laboratory-based testing sessions (day 1 and day 29), participants attended the laboratory before 8.00 a.m. after having consumed a standardised breakfast of cereal and/or toast at home no later than an hour before arrival. They must have refrained from alcohol for 24 h and caffeine for 18 h. On arrival, on each day, participants completed the State-Trait Anxiety Inventory (STAI) state subscale and the computerised cognitive assessment (as per Figure 2), followed by measurements of blood pressure and heart rate. Immediately following this, they consumed their treatment for that day. Two further cognitive assessments (plus blood pressure and heart rate) identical to the pre-dose assessment commenced at 120 (approximately 11:00 a.m.) and 240 (approximately 02:00 p.m.) minutes post-dose; the latter was taken in order to take advantage of the natural decline in performance during the day. Participants were offered a standardised lunch at approximately 12:10 p.m. in order to remove the potential confound of hunger for the final cognitive assessment in the afternoon. Lunch comprised 1 cheese sandwich (Hovis soft white bread 2 slices, with 186 kcal, 1.4 g fat, 2.8 g sugar and 7 g protein; Sainsbury’s British Medium Grated Cheddar Cheese at 30 g, with 127 kcal, 10.5 g fat, <0.5 g sugar and 7.6 g protein; and Lurpack slightly salted spread at approximately 10 g, with 72 kcal, 8 g fat, <0.1 g sugar and <0.1 g protein), 1 packet of ready salted flavour crisps (Walkers 25-g bag, with 132 kcal, 8 g fat, 0.1 g sugar and 1.5 g protein) and 1 pot of custard (Ambrosia 125-g pot, with 124 kcal, 3.5 g fat, 14.3 g sugar and 3.6 g protein). This lunch was optional (as long as non/consumption of components was the same for both visits) to avoid the potentially more disruptive effects of eating items which were unpalatable to participants.

Additionally, on day 25, participants came into the lab and completed the PRVP task as described above. At baseline on day 29, participants were prompted to recall the locations and actions from this PRVP task again but without the prompt of the video.

Figure 3 and Figure 4 depict the laboratory-based testing session timeline and chronic study overview, respectively.

### 2.5. Statistics

An *A Priori* G*Power [22] calculation determined that, to achieve a medium effect size (Cohen’s f = 0.15) with a minimum power of 0.8 utilizing the following analysis plan, a sample size of *n* = 86 would be required, which we rounded up to *n* = 90.

All of the below analyses were conducted with IBM SPSS Statistics 25 and were first investigated for normality and baseline differences between treatment groups. Unless reported, no baseline differences were detected. The confidence intervals for all analyses are set at 95% and, if post hoc analyses were required, these were student’s t-tests.

The COMPASS, Bond–Lader and blood pressure data were then analysed for acute effects within day 1 and day 29 as well as pure chronic effects within day 29:

1. Acute effects within day 1 and 29

Here, the post-dose performance for each day was changed from its own baseline and two-way, repeated measures ANOVAs were conducted with “treatment” as a fixed factor:

2. Pure chronic effects within day 29

Here, pre-dose and the two post-dose assessments on day 29 were changed from the day 1 baseline and two-way, repeated measures ANOVAs were conducted using “treatment” as a fixed factor.

The PRVP data was analysed via four methods:

1. Day 25 data: A univariate ANOVA with “treatment” as a fixed factor was used to analyse performance following 25 days (±3 days) of treatment. One full mark was awarded for correctly remembering the location with its action (this is defined as the “original scoring”).

2. Day 29 data (original scoring): See 1.

3. Day 29 data (lenient scoring): It was considered that the original scoring described above may be too conservative to utilize on performance captured 4 days (±3 days) after having encoded the location/action list, and therefore, a further analysis applying more lenient scoring was also undertaken. Here, 1 mark was awarded for all correctly remembered locations and all correctly remembered actions and a univariate ANOVA with treatment as a fixed factor compared this score between treatments.

4. “Decay” score: the score on day 29 was subtracted from the day 25 performance (original scoring method was used for both) to create a “decay” score, defining how much memory had degraded across the 4 days (±3 days), and a univariate ANOVA with treatment as a fixed factor compared this decay score between treatments.

The STAI mood data were analysed via two approaches. Firstly, the trait anxiety scores (only one value for each participant collected pre-dose) were analysed via univariate ANOVA, with treatment as a fixed factor, to determine whether the two groups had any intrinsic differences in anxiety. Secondly, to determine the effects of treatment on state anxiety scores, change scores (day 29 minus day 1) were compared between treatments via repeated measures ANOVA with treatment as a between-subjects factor.

## 3. Results

### 3.1. Compliance, Treatment Guess and Adverse Events

Mean compliance was 101%, with compliances ranging 83–121%.

A chi-square analysis showed that participants were able to subjectively detect that they were in the active condition (69% guessed correctly): X(1) = 5.79, *p* = 0.02.

Over the course of the study, 44 adverse events presented which could possibly be related to the study treatment (10 placebo and 34 sage treatments). These comprised:Muscular/bodily pain/injury: 9Cold/flu symptoms: 5Headache/migraine: 41 (individual reports from 14 participants)

All participants reported thinking that their adverse events were *not* related to supplement use, and symptoms in all cases resolved during the course of the study. All adverse events were reported as “mild” or “moderate” apart from 2 participants in the sage condition who reported headaches as severe. However, one of them also reported flu/cold-like symptoms.

### 3.2. Blood Pressure

No pure chronic effects of treatment on blood pressure were observed, but an acute interaction between “treatment × repetition” on heart rate was observed on day 1: F(1,93) = 4.15, *p* = 0.04. However, post hoc interrogation revealed no significant differences at either post-dose time point.

### 3.3. Prospective Remembering Video Procedure (PRVP)

No significant effects were observed on any of the four analysis approaches.

### 3.4. Mood

No significant effects were observed on state or trait anxiety as indexed by the STAI. With regards to the Bond–Lader mood scales, the only effect involving treatment was a single acute trend towards significance for “treatment × repetition” on day 29 for contentment, F(1,92) = 3.21, *p* = 0.08, but no significant effects at any repetition was observed in the post hoc comparisons.

### 3.5. COMPASS Tasks

Because the COMPASS task outcomes comprise both acute and chronic effects and they are more abundant than the above outcomes, they are separated here into first acute and then chronic effects for clarity. (See Table 2 for cognitive task scores in comparison with baseline and Table 3 (which also includes trends towards significance although these are reported more fully in Appendix A) for a full summary of the cognitive effects of treatment).

#### 3.5.1. Acute Effects

**Numeric working memory accuracy** evinced a significant acute main effect of “treatment” in favour of sage within day 29; F(1,92) = 4.87, *p* = 0.03. A significant “treatment × repetition” in the same direction was also observed on day 29, F(1,92) = 4.21, *p* = 0.04, and post-hoc ANOVAs revealed that this was influenced by post-dose repetition 1, F(1,93) = 8.4, *p* = 0.005, as no significant effect was observed at post-dose repetition 2, F(1,93) = 0.50, *p* = 0.48. On the **Corsi blocks span score**, a significant baseline difference was detected on day 1, where placebo participants had a significantly higher span score (mean 6.06) than those in the sage condition (mean 5.65), F(1, 88) = 4.49, *p* = 0.04. A significant acute effect of “treatment” was then observed on day 1, F(1, 92) = 4.2, *p* = 0.04 (where span score was higher for sage compared to the placebo) and on day 29, F(1, 92) = 10.58, *p* = 0.002. Again, span score was higher for sage versus placebo. (See Figure 5).

#### 3.5.2. Chronic Effects

**Numeric working memory accuracy** evinced a significant pure chronic interaction between “treatment × repetition” on day 29 in favour of sage, F(2, 184) = 4.49, *p* = 0.01, with post-hoc ANOVAs revealing no significant effects at pre-dose, F(1,93) = 0.80, *p* = 0.37; a trend towards significance at post-dose repetition 1, F(1,93) = 3.1, *p* = 0.08; and no significant difference at post-dose repetition 2, F(1,93) = 0.07, *p* = 0.79. (See Figure 6 for acute effect of “treatment” on day 29 and Figure 7 for acute and pure chronic interactions between “treatment × repetition” on day 29). On **Corsi blocks span score**, a significant pure chronic interaction between “treatment × repetition” was observed on day 29, F(2,184) = 7.15, *p* = 0.001. Post hoc ANOVAs revealed that, whilst there was no effect at the pre-dose time point, F(1,93) = 0.36, *p* = 0.55, sage was trending towards better performance at post-dose repetition 1, F(1,93) = 3.4, *p* = 0.07, and this reached significance at post-dose repetition 2, F(1,93) = 6.2, *p* = 0.02. (See Figure 8). **On name-to-face recall accuracy,** a significant pure chronic main effect of “treatment” was observed on day 29, F (1,92) = 4.98, *p* = 0.03, where accuracy was better for sage compared to the placebo. (See Figure 9).

Of the 5 global cognitive domains that can be derived from combining COMPASS task outcome measures, 2 evinced significant effects pertaining to “treatment”.

#### 3.5.3. Working Memory “Accuracy”

A significant acute effect of treatment was seen on day 1, F(1,93) = 7.8, *p* = 0.006, and a trend towards significance was seen on day 29, F(1,93) = 3.49, *p* = 0.07. On both days, accuracy was reduced in the placebo condition and sage attenuated this. (See Figure 10).

A pure chronic trend towards significance was observed for “treatment” on day 29, F(1,93) = 3.50, *p* = 0.07, and a significant interaction between “treatment × repetition” was also seen, F(1,93) = 3.7, *p* = 0.03. Here, accuracy reduced from baseline in the placebo condition at both post-dose repetition 1 and post-dose repetition 2, whereas it was increased in the sage group. (See Figure 11).

#### 3.5.4. Overall “Accuracy”

Acutely, a trend towards significance was observed for “treatment”, F(1,93) = 3.0, *p* = 0.09, and here, we see a similar pattern to the above working memory accuracy findings; on both days, accuracy was reduced in the placebo participants, and this was attenuated in the sage participants. (See Figure 12).

This acute effect on day 29 was supported by a trend towards significance for a pure chronic effect of “treatment” on day 29 in favour of sage, F(1,93) = 3.0, *p* = 0.08. (See Figure 13).

Due to space constraints, tables depicting means and standard deviations for all trial outcomes can be found in the online Appendix A. Here, you can also find the statistical analysis tables for all significant and non-significant outcomes.

## 4. Discussion

The results here demonstrate some clear, additive benefits of sage to individual task performance as well as an attenuation of natural declines in performance; with the effects clearly isolated to the accuracy and working memory performance cognitive domains. The most convincing effects of sage are seen on day 29 and the existence of pure chronic effects on these measures suggests that these effects are the result of a cumulative effect of sage consumed over 29 days.

As an example, span score on the Corsi blocks task was significantly better in the sage condition acutely on day 1 (*p* = 0.04) and more so on day 29 (*p* = 0.002) and the pure chronic effect here also (*p* = 0.001) reinforces that this is likely due to a cumulative effect of sage over 29 days. Improved accuracy on the numeric working memory task was also seen on day 29 following sage (“treatment”, *p* = 0.03, and “treatment × repetition”, *p* = 0.04), and these findings were reinforced by pure chronic effects on day 29 too (*p* = 0.01). One outcome measure evinced only a pure chronic effect on day 29 in response to sage, accuracy on the name-to-face recall task (*p* = 0.03), and without an acute effect within day 29. We would argue that this supports the use of the analysis plan used here, where both acute effects (comparing post-treatment performance to pre-treatment performance on that same day) and pure chronic effects (comparing post-treatment performance on day 29 to pre-treatment performance on day 1) are investigated. This approach disentangles these effects, where taking treatment on day 29 does not have a significant acute effect in itself and is only revealed when analysis views this in relation to being the 29th dose of the intervention.

Taken together, it is clear that the benefits of sage are focused on the accuracy of performance (and on tasks with working memory as their key cognitive domain) in particular and so it is not surprising that, when the global cognitive domains were analysed, it was the accuracy of working memory factor which yielded a significant acute result on day 1 (*p* = 0.006) and a pure chronic effect on day 29 (*p* = 0.03). It is worth noting here that some of the above effects were significant main effects of treatment and/or interactions between treatment and repetition, and regarding the latter, it was always the case that one, not both, of the post-dose repetitions evinced significant differences between treatments. This likely speaks to the impact of individual variability in any number of factors, e.g., attentional focus and/or the pharmacokinetics of nutritional interventions. This justifies the use of protocols like that employed here, where cognition is assessed over relatively long periods of time and repeated multiple times. This not only amplifies the power of analysing these outcomes but also extends the window of opportunity for cognitive assessment, capturing effects in those who achieve plasma levels much sooner or later than others or during a window of the day where responsiveness to the intervention is higher. Future investigation into this potential pharmacokinetic variability would be insightful and would likely confirm that variable effects at different time points is the result of fluctuating plasma levels of the intervention and/or time of day effects.

One key aim of this study was to mimic the persistent attentional and memory demands elicited in everyday life across various media, and so, it is particularly interesting that some of the above benefits of sage can be viewed as an attenuation of naturally depleting performance. Both span score on the Corsi blocks task and accuracy on the working memory accuracy global cognitive domain were depleted in the placebo condition (i.e., performance reduced from pre- to post-dose), but sage was able to prevent this depletion from being as severe. Why the performance is so depleted in the placebo group is an interesting point. A recent review [23] demonstrated that both time of day (namely later in the day) and lack of movement were two key predictors of poorer cognition, alongside mood and motivation. As this trial required participants to be sedentary for large periods of time during cognitive task completion and the fact that this continued into the afternoon might simply suggest that sitting for so long was the cause. Whilst this does represent many real-world scenarios, e.g., where we may need to sit and focus on a task for large portions of the day, it would be interesting to see if sage could outperform the placebo in situations where participants were more active.

Historically, the cognitive effects of sage seemed to be isolated to improved memory (specifically recall) and attention alongside increased alertness, calmness, contentment and reduced mental fatigue [2,4,5,10,11], which obviously presents a partial deviation from the results seen here. Here, we saw no changes in mood, and whilst memory does seem to be the prevailing cognitive domain affected, here, it is working memory rather than recall, as seen in the aforementioned original trials conducted by this lab. When interrogating the differences/commonalities between these historical trials and the current trial, it is important to note that the cognitive tasks used were delivered via a platform similar to COMPASS (and so the tasks and their completion requirements were very close) and, in some trials, the testing time was comparable. The age of participants in the current trial were also investigated in these previous trials; albeit no one trial covered the whole range used here. As such, the only notable deviances which might explain the different results reported here are the increased power (*n* = 94 here compared to Ns ranging from the 20 s to 30 s previously) and that the original trials utilized *Salvia officinalis/Salvia lavandulaefolia* in isolation in the form of an essential oil compared to the use of a *Salvia* combination and the addition of the naturally co-occurring polyphenols in a dried-leaf form used here.

The polyphenols identified in *Salvia officinalis* and *Salvia lavandulaefolia* include rosmarinic acid, methyl carnosate, caffeic acid, luteolin 7–0-glucoside, luteolin, apigenin and hispidulin. The former also synthesizes salvianolic acid derivatives of rosmarinic acid. These polyphenols are present alongside terpenes like α- and β-pinene, 1,8-cineole, camphor, geraniol, borneol and camphene (see [1] for review), and therefore, the individual and symbiotic effects of individual phytochemicals cannot be attributed to the effects seen here. One might consider this attribution unimportant anyway considering that, when consuming this specific intervention, similar products and indeed dietary sage, it would be as part of a phytochemical cocktail and not isolated compounds. This may explain why no reports exist on the effects of the abovementioned individual phenolic acid and terpene constituents on memory function in humans.

The hypothesis of differential effects in response to this altered intervention is strengthened by the preclinical trial described in the Introduction section which was conducted in rodent models using the combination of sage used here (Cognivia™) [12]. This study demonstrated a synergistic effect of the combination in comparison with a *Salvia officinalis* aqueous extract or *Salvia lavandulaefolia* essential oil alone, and these findings were particularly apparent following 2 weeks of intervention. Moreover, the effects of this combination in rodents mirrored those seen here in humans, focused as they were on learning and spatial memory functions as observed in the Morris water maze.

This recent preclinical trial might shed some light on why this combination would preferentially target working memory and accuracy performance, specifically its finding that calcium/calmodulin-dependent protein kinase II (CaMKII) expression was increased. CaMKII is a key enzyme in all brain regions, recruited around synapses, and is implicated in several neurotransmitter metabolic pathways, including serotonin, with relevant activity in postsynaptic signal propagation (ion channel modulation and calcium homeostasis), synaptogenesis and synaptic plasticity. In particular, CaMKII-induced synaptic strengthening leads to long-term polarisation (via actin remodelling), a major biochemical pathway supporting working memory and cognitive processes such as learning, reasoning and interpretation [13,14,15,16].

As supplementary points, it is worth noting that the effects of treatment were observed neither on mood (which is likely explained by the abovementioned difference in the active ingredients between this and historic trials) nor on the PRVP (video) task. The absence of effects on the video task may be as a result of this task hitherto proving sensitivity only to the effects of insults to prospective memory rather than any additive benefits to performance from some intervention. This does not mean that this task or the domain of prospective memory should not be taken forward in future intervention trials but rather that its sensitivity may be more subtle than other tasks/tools that have been long-validated in this area.

Secondly, this study did not restrict or assess the diet of participants prior to or during the study. The aim of this trial was to recruit a random cohort of participants and, with little to no research to indicate that particular dietary approaches would impact the effects of a salvia intervention, the inclusion of any kind of dietary measurement/control never entered considerations for the methodology employed here. Without question, assessing the interaction between diet and other lifestyle factors on all interventions/supplements is an interesting point. However, due to the incalculable variability within and across participants when it comes to diet and lifestyle, if this is not going to be a primary outcome measure of a trial design (with the cohort size to sustain it—one which we certainly did not have here), then a crude assessment of diet and/or a small sample size likely would not provide any great insight and would be more likely to confuse interpretation of other outcome measures.

Thirdly, the decision to supplement for 29 days was somewhat arbitrary in the face of no evidence on the cumulative effects of *Salvia* in humans; as such, a month was deemed an appropriate and relatively easily achievable (confirmed by the mean compliance of 101%) supplementation period for this initial investigation. Practically, chronic supplementation studies must allow a window of time during which participants can return for the final visit, and here, +/− 3 days was decided on as an appropriate range as this would incorporate when the return visit fell on the weekend. Ultimately, however, 87 of the 94 (93%) analysed participants were supplemented between 28–30 days, and again, as no data suggests that supplementary effects of sage would differ outside of this range for the remaining 7% of participants, one cannot say whether this would have impacted the results.

Finally, it should not go unmentioned that participants were able to subjectively detect that they had been in the active intervention group (i.e., 69% correctly guessed this when asked at their final visit), and it would be remiss to think that this could not have impacted the results via participant expectancy effects. Some of this awareness was due the perception of a herbal taste during reflux, but some was based on an identification of a change in mental ability. It could be argued that the ability to subjectively detect this in oneself is a positive outcome, especially as it coincided with statistically detectable improvements in mental ability and speaks to the potency of this extract in being able to induce such changes. Whilst very little can be done to remove the subjective perception of increased mental ability (really, the only option here would be to incorporate an active control into the design), certainly, future studies could improve on the formulation or encapsulation method of the intervention to conceal the herbal taste.

In conclusion, we have observed a consistent significant benefit of a sage combination intervention (Cognivia™) in healthy adult humans on working memory and accuracy of performance cognitive domains. This significant activity was observed both acutely (after just 2 h following consumption) and chronically (after 29 days of administration). The pattern and magnitude of significance points towards an increase in product efficacy over the administration period and, taken together, suggests that future trials should focus on disentangling the working and spatial memory effects of this intervention in humans with an extended timeframe of perhaps several months. Validating the CaMKII mechanism in humans would also be advantageous.

## Figures and Tables

**Figure 1 nutrients-13-00218-f001:**
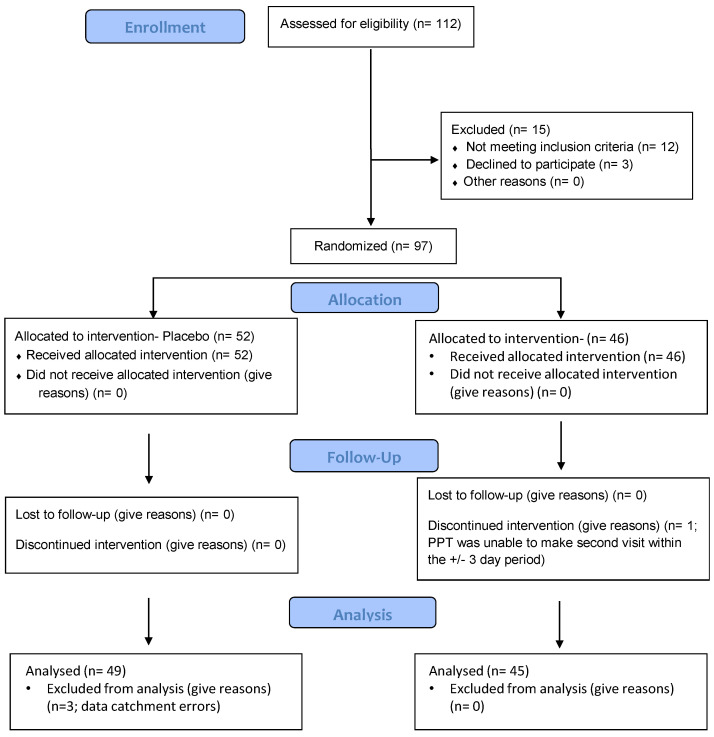
Participant disposition diagram. From the 112 potential volunteers enrolled, 97 were randomised and allocated to receive either treatment A (placebo; *n* = 52) or treatment B (sage; *n* = 46). With 1 participant lost to follow up in the latter group and 3 excluded from analysis due to data catchment errors in the former group, the final total of data sets available for analysis were *n* = 49 in the placebo condition and *n* = 45 in the sage condition.

**Figure 2 nutrients-13-00218-f002:**
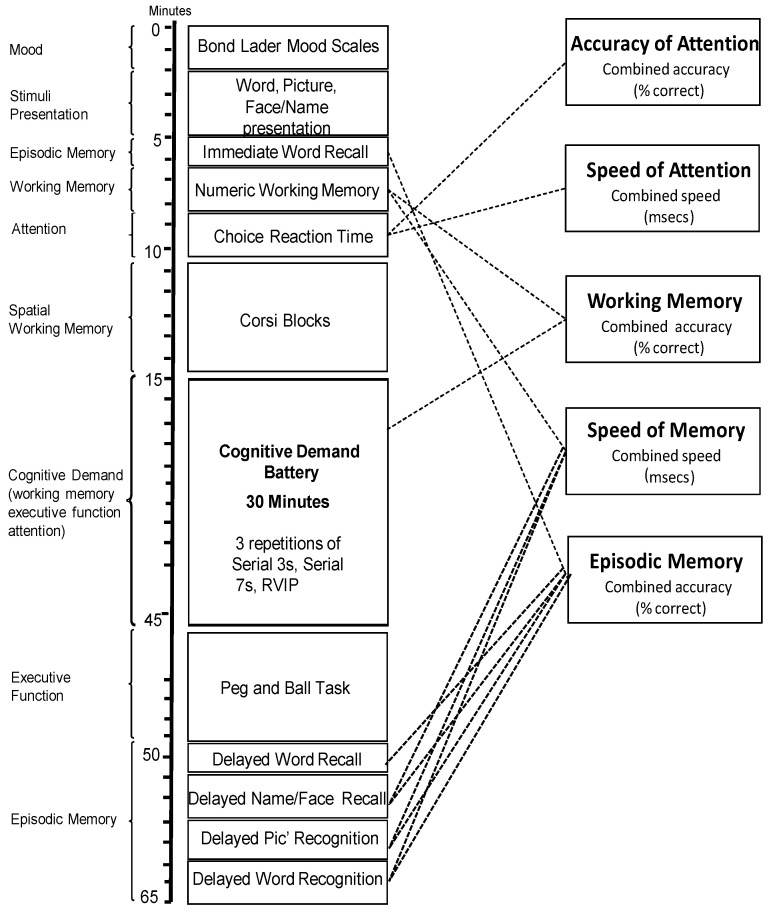
The running order of the individual cognitive assessments. The tasks are shown in order of completion with approximate timings. On the left, the “cognitive domain” assessed by the task is shown, and the boxes to the right show global measures into which data from several tasks can be collapsed.

**Figure 3 nutrients-13-00218-f003:**
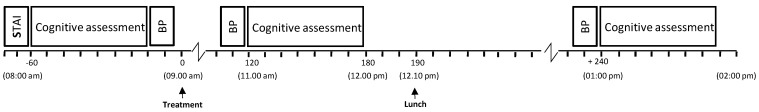
Testing session timeline on both the acute and chronic visits. Participants completed the state subscale of the State Trait Anxiety Inventory (STAI) followed by a full cognitive assessment pre-dose and at 120 and 240 min post-dose. Blood pressure and heart rate were recorded after each cognitive assessment. Lunch was provided at approximately 12:10 p.m.

**Figure 4 nutrients-13-00218-f004:**
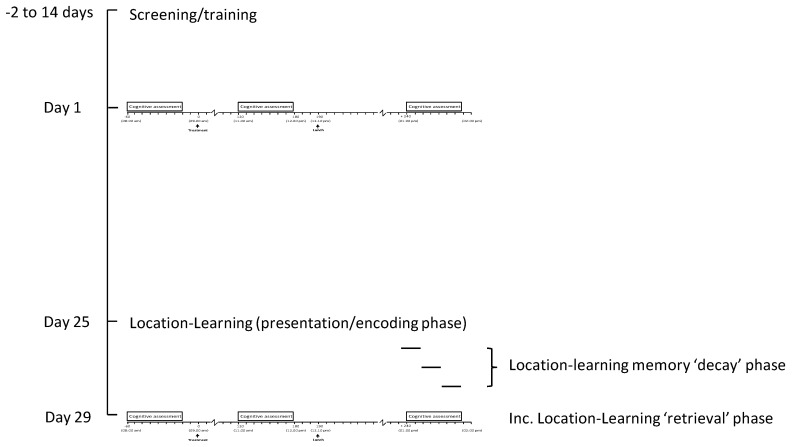
Overall trial diagram. The diagram includes screening/training and the main acute (day 1) and chronic (day 29) lab visit sessions. Participants returned to the lab on day 25 to first complete the location-learning task. The 3 intervening days between this and day 29 when they again completed the location-learning task acted as a “decay” phase for these encoded memories.

**Figure 5 nutrients-13-00218-f005:**
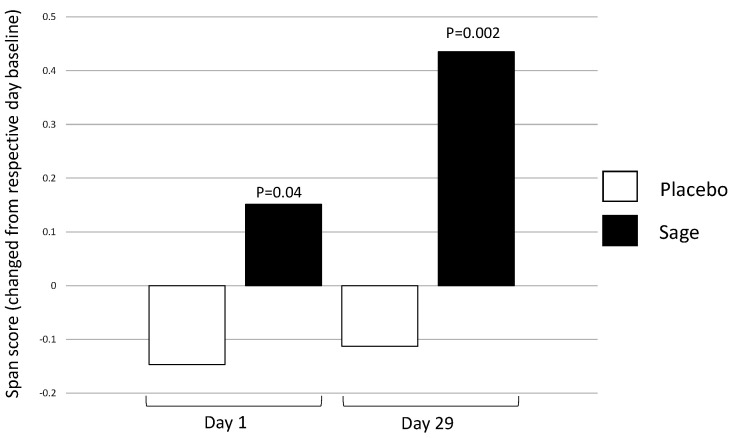
Acute effect of treatment on Corsi blocks span score on day 1 and day 29: a significant main acute effect of treatment was observed on both day 1 (*p* = 0.04) and day 29 (*p* = 0.002) when post-dose performance was compared to the pre-dose performance on the same day.

**Figure 6 nutrients-13-00218-f006:**
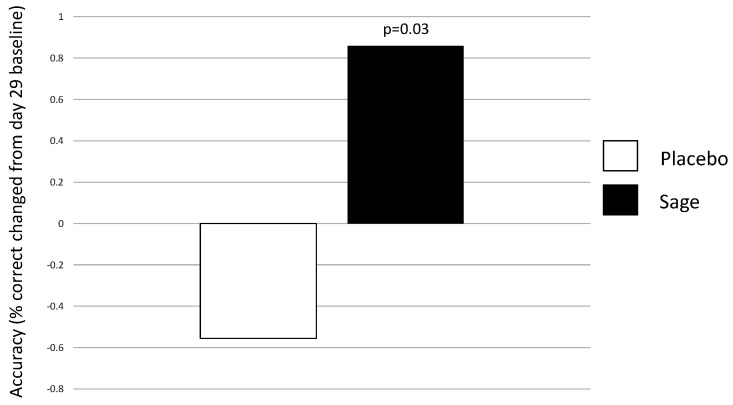
Acute main effect of treatment on day 29 on numeric working memory accuracy: a significant acute main effect of treatment was observed on day 29 (*p* = 0.03). Here, percentage change from baseline (day 29 pre-dose) accuracy was higher in the sage condition than the placebo.

**Figure 7 nutrients-13-00218-f007:**
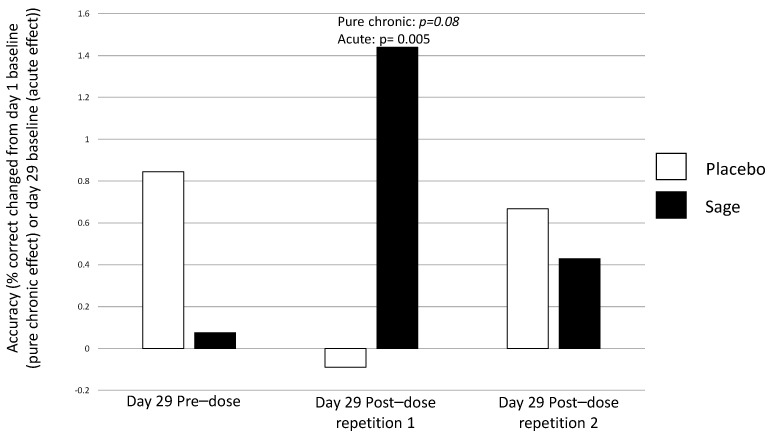
Acute (day 29) and pure chronic interaction between treatment × repetition on numeric working memory accuracy: both an acute interaction (*p* = 0.04) and pure chronic interaction (*p* = 0.01) between treatment × repetition was observed on day 29. Both revealed that this effect took place at post-dose repetition 1; the acute effect compares this post-dose repetition to the day 29 pre-dose performance (*p* = 0.005) and the pure chronic effect compares this post-dose repetition to the day 1 baseline (*p* = 0.08).

**Figure 8 nutrients-13-00218-f008:**
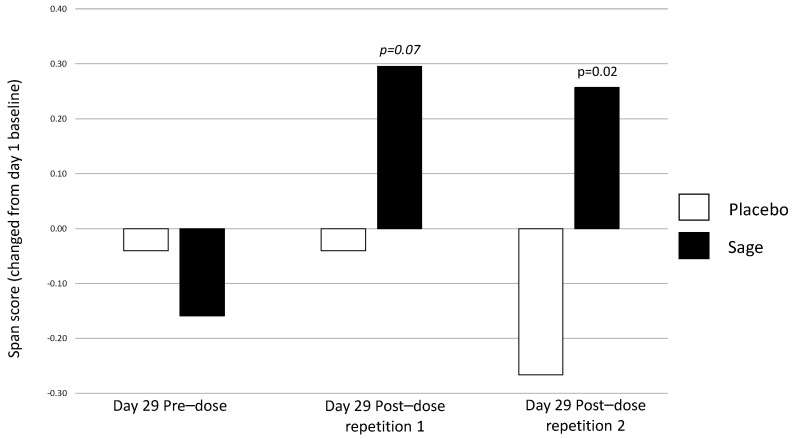
Pure chronic interaction between treatment × repetition on Corsi blocks span score: a significant pure chronic interaction between treatment × repetition (*p* = 0.001) was observed, and here, when compared to day 1 baseline performance, the post-dose repetition 1 span score trended (*p* = 0.07) towards being significantly better in the sage condition than the placebo and was significantly better at post-dose repetition 2 (*p* = 0.02).

**Figure 9 nutrients-13-00218-f009:**
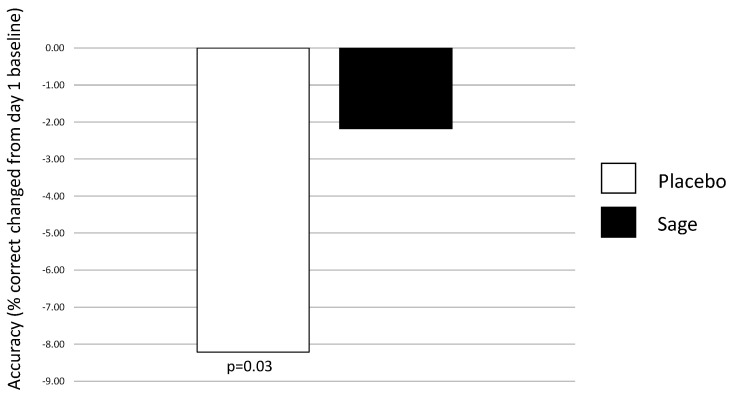
Pure chronic main effect of treatment on day 29 for name-to-face recall accuracy: a significant acute main effect of treatment (*p* = 0.03) was observed on day 29 where accuracy was significantly lower (as compared to pre-dose performance on day 1) in the placebo condition as compared to the sage condition.

**Figure 10 nutrients-13-00218-f010:**
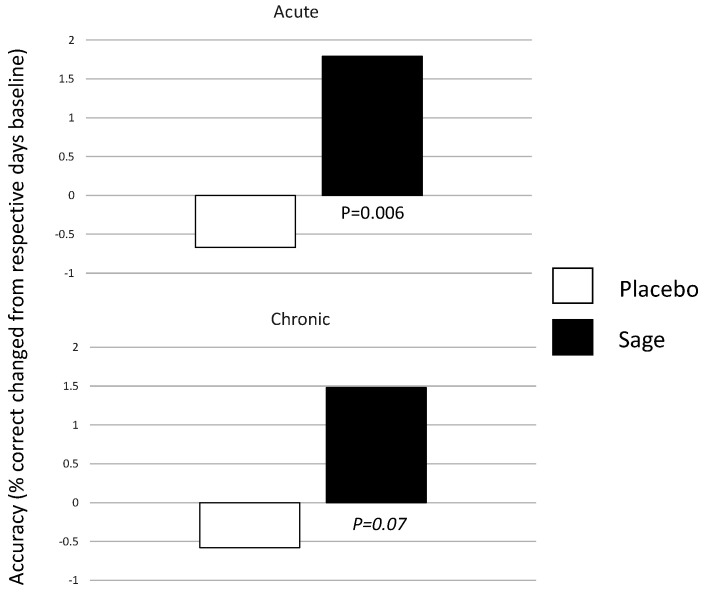
Acute effect of treatment on working memory accuracy on day 1 (top) and day 29 (bottom): the effects of treatment were observed acutely on day 1 (*p* = 0.006), and this was trending towards significance for day 29 also (*p* = 0.07). In both cases, accuracy was higher (as compared to their own days’ pre-dose performance) in the sage condition as compared to the placebo condition.

**Figure 11 nutrients-13-00218-f011:**
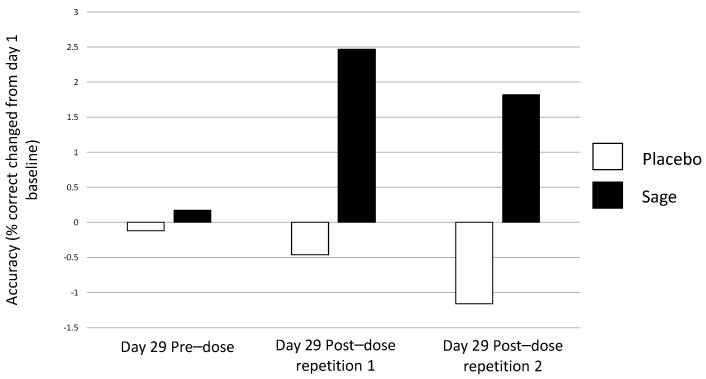
Pure chronic interaction between treatment × repetition on working memory accuracy on day 29: a significant pure chronic interaction between treatment × repetition (*p* = 0.03) was observed on day 29, where accuracy was significantly better (as compared to baseline performance on day 1) in the sage condition at both post-dose repetition 1 and 2 as compared to the placebo condition.

**Figure 12 nutrients-13-00218-f012:**
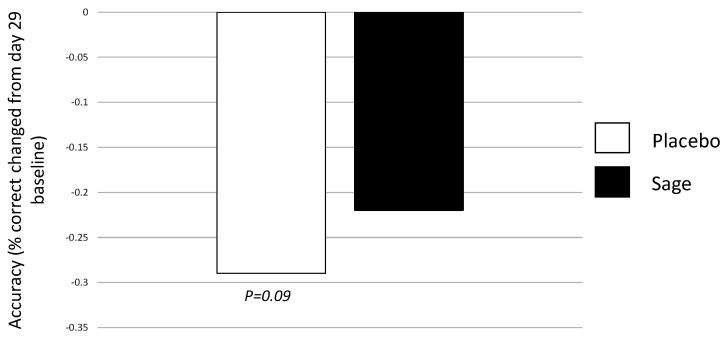
Acute effect of treatment on day 29 for overall accuracy: a trend towards a significant main effect of treatment (*p =* 0.09) was observed for overall accuracy on day 29, where accuracy was poorer (as compared to pre-dose performance on the same day, day 29) in the placebo condition as compared to the sage condition.

**Figure 13 nutrients-13-00218-f013:**
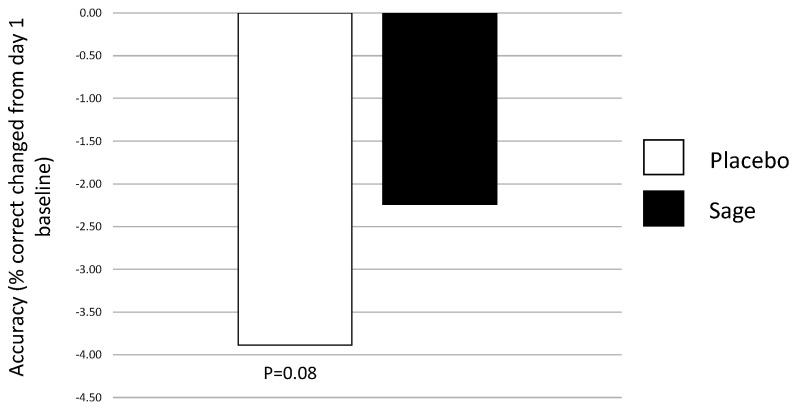
Pure chronic effect of treatment on day 29 for overall accuracy: a trend towards a significant pure chronic effect of treatment (*p* = 0.08) was observed, where accuracy was significantly poorer (as compared to baseline performance on day 1) in the placebo condition as compared to the sage condition.

**Table 1 nutrients-13-00218-t001:** Participant demographics for the placebo and sage conditions.

Measure	Placebo	Sage	Total
*n*	49	45	94
Average years of age (SD)	44.7 (8.4)	43 (8.9)	43.9 (8.6)
Male/Female	11/38	13/32	25/69
Left-/Right-handed	3/46	3/42	6/88
Average years in education (SD)	17 (3.5)	16.5 (3.6)	16.7 (3.6)
Average Body Mass Index (SD)	24.8 (3.3)	26.1 (3.6)	25.4 (3.5)

It outlines the demographic composition of participants as assessed at screening/training in both the placebo and sage conditions as well as an average across both groups. No significant differences between treatment groups were observed on these demographic features.

**Table 2 nutrients-13-00218-t002:** Cognitive task scores in comparison with baseline.

	Treatment Condition	Day 1						Day 29					
		Baseline		Post-Dose 1		Post-Dose 2		Pre-Dose		Post-Dose 1		Post-Dose 2	
				Difference from Baseline		Difference from Baseline		Difference from Baseline		Difference from Baseline		Difference from Baseline	
		Mean	SD	Mean	SD	Mean	SD	Mean	SD	Mean	SD	Mean	SD
Numeric Working Memory Accuracy	Placebo	95.29	4.66	0.64	4.13	0.67	5.74	1.33	4.77	−0.09	4	0.67	4.71
	600 mg Sage	95.51	4.04	0.35	3.94	0.88	3.87	0.08	3.92	1.44	4.38	0.43	3.88
Corsi Blocks Span	Placebo	6.07	0.82	−0.13	0.8	−0.16	0.67	−0.04	0.74	−0.04	0.72	−0.27	0.98
	600 mg Sage	5.67	1	0.18	0.96	0.12	0.9	−0.16	1.15	0.3	1.03	0.26	1.07
Name-to-face-Recall Accuracy	Placebo	65.42	16.86	−9.1	13.14	−8.83	17.59	−2.5	15.77	−6.75	14.7	−9.67	16.78
	600 mg Sage	63.35	18.63	−4.26	19.26	−5.97	17.58	4.17	19.06	−0.19	19.15	−4.17	17.66

It shows the scores on COMPASS tasks which are implicated in significant differences between groups. Day 1 baseline values denote raw means, and subsequent columns denote means and standard deviations (SD)s that changed from this baseline.

**Table 3 nutrients-13-00218-t003:** Summary of cognitive effects of treatment.

Task	Outcome Measure	Acute		Pure Chronic
		Day 1	Day 29	
Corsi Blocks	Span Score	treatment *p* = 0.04	treatment *p* = 0.002	treatment × repetition *p* = 0.001
Numeric Working Memory	Accuracy	*n*/A	treatment *p* = 0.03 and treatment × repetition *p* = 0.04	treatment × repetition *p* = 0.01
Peg and Ball	Thinking Time	treatment × repetition *p* = 0.08	N/A	N/A
Name-To-Face Recall	Reaction Time	N/A	treatment *p* = 0.06	treatment × repetition *p* = 0.07
	Accuracy	N/A	N/A	treatment *p* = 0.03
Serial 3 Subtractions	Errors	N/A	treatment × repetition *p* = 0.06	treatment × repetition *p* = 0.06
	Total	treatment × repetition *p* = 0.07	N/A	N/A
	Accuracy	treatment × repetition *p* = 0.07	N/A	N/A
Serial 7 Subtractions	Accuracy	treatment × repetition *p* = 0.08	N/A	N/A
Working Memory	Accuracy	treatment *p* = 0.006	treatment *p* = 0.07	treatment *p* = 0.07 and treatment × repetition *p* = 0.03
Overall Accuracy	Accuracy	treatment *p* = 0.09	treatment *p* = 0.09	treatment *p* = 0.08

It shows the outcome variables from COMPASS which evinced significant effects involving treatment as a factor. For completeness, this table also depicts the trends towards significance (latter in italics) involving treatment as a factor. These were not included in the above textural report of the results for brevity, but the authors believe that they do contribute to the full picture of results and so the full report of trends towards significance can be found in the online Appendix A. Outcomes defined within a thick black border denote the global cognitive domains. N/A = not applicable (i.e., no significant effect containing “treatment” was observed here).

## Data Availability

The data presented in this study are available on request from the corresponding author. The data are not publicly available as they pertain to a proprietorial product.

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
