# Peer review of "The Acute and Chronic Cognitive Effects of a Sage Extract: A Randomized, Placebo Controlled Study in Healthy Humans"

_nutrients, 2021, doi:10.3390/nu13010218_

Round 1

Reviewer 1 Report

The manuscript entitled “The acute and chronic cognitive effects of a sage extract: a randomized, placebo controlled study in healthy humans” presents an interesting issue, however it required some corrections.  

ABSTRACT:

  • Line 14 - word “Background:” is redundant (structured abstracts, but without headings)
  • Line 20 - word “Objective:” is redundant (structured abstracts, but without headings)
  • Line 22 - word “Design:” is redundant (structured abstracts, but without headings)
  • Line 25 - word “Results:” is redundant (structured abstracts, but without headings)
  • Line 29 - word “Conclusions:” is redundant (structured abstracts, but without headings)

INTRODUCTION:

  • Line 35-41 – this paragraph required references
  • Line 51 – “Wightman et al. In prep]” – please remove non-reviewed manuscript

MATERIALS AND METHODS:

  • Taking into account the in group with placebo and group with sage have differing number of participants (55 vs. 44). It would be recommended for authors to applied the pair-matched design. The matched pairs design is an improvement over a completely randomized design.
  • How and where participants were recruited? Please specify.
  • Lines 93-94 – “had any pre-existing medical condition/illness which will impact taking part in the study” – please specify
  • Lines 94-95 – “were currently taking prescription medications which would contraindicate with the study” – please specify
  • Please justify the cut for BMI (BMI outside of the range 18.5-30 kg/m2)
  • Lines 108-109 - For the research that involves human subjects the rules of the  Declaration of Helsinki of 1975 must be applied, including ethics commission approval. Please add the information about number of ethics commission approval (specific reference) - NOT department of Psychology approval.
  • Line 142 – please improved the reference style
  • Lines 143-145 – “Picture Presentation” more details information must be presented

RESULTS and DISCUSSION:

  • Line 393 – the title of the table must be corrected.
  • Line 396 – the title of the table must be corrected.
  • Tables – “and SD in italics alongside” there is no need to do it in italics – please just named the column.
  • The discussion section must be significantly improved. Authors should relate the findings to those of similar studies and point the differences and similarities between the studies. Authors should add the appropriate references in this section!

Minor comments:

  • Authors should follow the Instructions for authors while preparing their manuscript (!)

Reviewer 2 Report

RE: Review Nutrients-1027530

Comment to Authors

The Authors conducted a double blinded parallel randomized control trial in healthy humans to investigate the acute and chronic (29 days) effect on cognitive performance induced by a daily administration of COGNIVIATM, a combination of Salvia officinalis and Salvia lavandulaefolia extracts. Authors confirmed that the combination of two Salvia extracts has beneficial effects on working memory and accuracy of performance following both acute and chronic administration of COGNIVIATM.

The present study’s strengths are the notable number of subjects recruited for this RCT (n=94), the investigation of both acute (two post-administration assessments within day 1 and day 29) and chronic (between day 1 baseline and day 29) effects on cognitive functions, the assessment of the effects of a combination of two Salvia extracts, the administration of a battery of tasks which allows to measure both mood and cognitive performance and to assess specific cognitive domains (e.g. working memory).

However, although the experimental design is promising, the present study presents a number of weaknesses, which raised serious concerns.

  1. INTRODUCTION

A clear defined and original research question, supported by a strong and referenced state of art and rationale is missing.

What is the standing question of this work? The support to the hypothesized benefits in administrating both Salvia officinalis and Salvia lavandulaefolia extracts is provided by a single animal work done by one of the Author, who is affiliated with the company, which produces COGNIVIA TM.

Are the Authors aimed at filling some gap in the literature? Or they just want to replicate in humans their animal data?

Perhaps it would be interesting to indicate the natural phytochemical composition of these two plants extract: are they different?

Why COGNIVIATM combines Salvia officinalis polyphenols and Salvia lavandulaefolia terpenoids?

On what evidences the choice of this combination was based?

Is the concentration of these phytochemicals different in these 2 selected plants?

The Authors should for instance provide information on the effects of the administration of each type of Salvia. More specifically, why combining the polyphenols of one type and the terpenoid of the other type?

Has it been previously demonstrated the major effects of Salvia officinalis polyphenols and Salvia lavandulaefolia terpenoids separately on cognitive functions and mood?

  1. METHODS

Although a diagram about the participant allocation has been provided, some aspects of the methods might need some clarifications to fully appreciate methods and subsequently results.

2.1 Study design and participant.

Was the power analysis done a priori?

How was performed the randomization?

The trial was double-blinded: who was blind to the procedures?

There are some discrepancies between the numbers in the text and the numbers in the diagram. Please check.

Mean values regarding age, BMI, years in education of all 94 participants was provided, along with sex ratio. It would be interesting to indicate mean values and sex ratio for the two experimental groups, perhaps as a table, indicating possible significant differences among the groups.

Was the dietary pattern and type of food usually consumed assessed? This might influence the individual intake of polyphenols and terpenoids.

How the “healthy” condition was established? Could the Author specify what the “sanitary screening” consisted of?

2.2 Treatments.

The “Conflict of Interest” section is empty. Although the Authors do not provide any information on a possible conflict of interest, I am afraid this may exist, since one of the Authors is affiliated with Nexira SAS, which supports financially the study and produces Cognivia TM (https://www.nexira.com/brand/cognivia/). In this regard:

Is COGNIVIATM is it available in the market?

Is it a product formulated in the lab for previous and the present study?

It is stated that both the aqueous extract and the essential oil were characterized for their content in either polyphenols or terpenoids. Who performed the characterization? Is there a reference to a paper that can be consulted?

Was water the solvent used for aqueous extraction?

What was the composition of the placebo?

Furthermore, it is stated either placebo or COGNIVIATM was consumed every day for 29+/-3 days. Have some participants continued consumption less than 29 days?

At day 1 and day 29 administration of COGNIVIATM was at 9:00 am. Has this time of administration been followed throughout the 29 days? Have some differences emerged from the treatment diary?

2.3 Cognitive, mood and blood pressure assessment.

The reference (Wightman et al., 2015) cannot be found in the reference list, however it seems that the cited article reported only description of the Cognitive demand battery and not of the other tasks. Please check.

For acute assessments, tasks were repeated at baseline, after 120 min and 240 min. Is it not clear whether the images, words, numeric sequences (etc.) were different for each assessment. Please specify.

It would be of interest for the readers to report some information of the method used to combine COMPASS task outcome measures to evaluate global cognitive domains.

2.4 Procedure.

Regarding the assessment of mood, there is a discrepancy between the scheme in fig.2 and the text. Blood pressure and Heart Rate measurements are not included in the scheme. Please do.

It was reported that offered standardized lunch was not consumed by some participants.

How many of them decline lunch offer?

Were they allowed to eat something else or did they perform tasks at 1 pm (2 assessment) in a fast state? This might affect their cognitive performance in the last assessment of the day.

2.5 Statistics.

This section should provide more information. Authors should specify whether the G power calculation was performed a priori o posteriori, whether data was assessed for normality.

For the acute and chronic effect, please specify whether a TWO-WAY ANOVA repeated measures was applied.

In results post-hoc tests were mentioned, but it was not specified the type of test applied.

What software was used?

The Authors refer to 3 age groups (which is reasonable considering the wide-range of ages of participants), but afterwards no reference to results by age group is provided. This is important since cognitive may be affected by age. Comparison cannot be drawn between young and aged people or it may be necessary to consider age as a covariate in data analysis.

  1. RESULTS

This section is hard to read because of the number of abbreviations used here (and in the Discussion) for the first time and discrepancies between tables, graphs and text.

Authors should present data (both in the text, in the graphs or tables) more appropriately. The analysis performed and the outcomes investigated appropriately should be clearly stated. In this actual state it is difficult to fully understand and appreciate the results and the statistical significances emerged.  As example of the found discrepancies: It is reported that 9 task outcomes variables analysed yielded a significant result associated to treatment. Referring to table 1 (as indicated by the authors) did not report these 9 significantly different outcomes

Acute and chronic effects are reported in the text mixed together, they should be reported separately. Possibly in a table.

Furthermore, it is reported that 69% of participants guessed correctly that were in the “active condition”.  In this case, I suspect that the blinded condition may not be fully achieved.

Should be interesting to know if the analysis of the outcomes has been performed also considering the age of participants.

  1. DISCUSSION

Results are not fully discussed within the existing literature. The Authors proposed that the differences observed between this study and the previous ones were due to the administration of the combined extracts, however this should be proven perhaps in a parallel study with single and combination of extracts.

Depletion of some task outcomes in placebo condition has been mentioned in the discussion, but not addressed in relation to literature.

In conclusion authors mentioned to the efficacy of COGNIVIA on working memory and accuracy of performance (are they referring to the accuracy as outcome measured to evaluate the cognitive domain “working memory”)?

SPECIFIC COMMENTS:

Line 55: reference 2 is an in vitro study, although in the text it is mentioned a human study.

Line 56: “the effect of a single dose…” what type of effect? On cognitive functions?

Line 57: “in the first of these study…” first based on what?

Line 58: reference 4 in the study it was administered dried leaf

Line 90: discrepancies between the text and the diagram: Placebo group 49 in the text and 52 in the diagram (50 analysed)

Line 142: Reference Wightman et al., 2015 is not in the format of the paper and it is missing in the reference list

Fig 2: discrepancy with the procedure in the text regarding assessment of mood. What does VAS stand for?

Line 212: Reference 17: PRVP was not assessed during hangover, but in the binge drinking context.

Line 346: does “PD” stand for post-dose?

Table 2: repetition*treatment? Or treatment*repetition (as it is throughout the text)?

Fig 10: bottom graph is the acute assessment at day 29?

Round 2

Reviewer 1 Report

I appreciate the great efforts that the authors have made in response to my questions and concerns. However, I have some additional comments:

  • In case of title of tables -  “Table 2. Cognitive task scores in comparison with baseline. Table shows scores on COMPASS tasks which are implicated in significant differences between groups. Day 1 Baseline values denote raw means (and SD in italics alongside) and subsequent columns denote means and SDs standard deviations (SD)s which are changed from this baseline”; “Table 3. Summary of cognitive effects of treatment. Table shows outcome variables from COMPASS which evinced significant effects involving treatment as a factor. For 461 completeness, this table also depicts those trends towards significance (latter in italics) involving treatment as a factor. These were not included in the above textural 462 reporting of the results for brevity but the authors believe that they do contribute to the full picture of results and so the full reporting of the trends towards significance 463 can be found in the online Supplementary Materials. Outcomes defined within thick black border denote Global Cognitive Domains.“ - The title should be corrected – there is some problem with formatting and the text is combined with the title.

Author Response

Apologies. Those pieces of text should have been presented below the table (not within the title) and I have now done this. Just let me know if it's not correct still and I can do that ASAP- you might get an out of office email reply but I am still checking my emails up to the 24th December.

Emma

Reviewer 2 Report

The Authors addressed many of the concerns expressed in the first revision and I appreciate the effort they put in improving the manuscript.

However, there are still some important points that need to be further clarified, as reported below.

INTRODUCTION

  1. Although the authors have implemented the introduction indicating the knowledge gap in this field of interest, it has not been stated a clearly defined and original research question yet. Considering the state of art, please define and report the aim of the study.
  2. The Author answered the questions on the phytochemical composition of these the two plants extract and the choice of combining Salvia officinalispolyphenols and Salvia lavandulaefolia terpenoids, however they should check whether the references are correct (line 60, ref 4 they tested salvia officinalis, not lavandulaefolia).
  3. It would be appreciated to specify which specie of salvia was used in the cited studies for a better understanding (line 56 and 57). Sentence from line 57 to 59, please insert a reference.
  4. It was appreciated the note explaining the chemical composition of the two species. However, since it is an important point it should be reported in the introduction text with appropriated references.
  5. The note reports only information about the terpenoid fraction (1,8 cineolo and canfora), but what about the polyphenols fractions? Is there any information in the literature?
  6. Furthermore, what are the effect of canfora (which is significantly different between the two sage species) on cognitive functions? It might be an interesting point of discussion.
  7. The Authors addressed the question on why combining the polyphenols of one type and the terpenoid of the other type properly. They stated they are testing just the salvias and not isolated phytochemicals. However, the phytochemicals are components of the extracts they are testing. Therefore, the effects on cognition of these phytochemicals should be taken into account and argued in the discussion with associated references. This might provide support to the results and conclusions.
  8. I agree it is difficult to specifically state which terpenes and phenols are producing these effects in this study. However, the composition of phytochemicals of the two sage species and differences should be reported in the text and perhaps, it would be interesting to discuss results and conclusion considering the effect of terpenes and phenols (in vitro, in vivo studies?) and explain the results and conclusions.

METHODS

  1. The Author have included the information about the method used to randomize the groups. Was the randomization done by extraction? The groups were homogeneous after simple randomization?
  2. There are still some discrepancies between the numbers in the text and the numbers in the diagram. Please check Line 102: “The number of participants in the placebo condition was 49 and 45 participants consumed the active intervention.”, but in the Figure 1 in the last box on the left “Analysed (n=50)”. Please check also in figure 1 description.
  3. Table 1 should also indicate p values. Please indicate continue variable as mean ±SD and categorical variables as n and %.
  4. Perhaps my request to provide information on the dietary pattern and type of food usually consumed by the subjects has not been fully understood. Although we do understand the difficulties of controlling diet and even more to prescribe a nutritional protocol, diet is anyway an important factor to consider in this kind of studies. Therefore, it would have been interesting to know participants’ nutritional habits (by means of a Food Frequency Questionnaire or a 7-day nutritional diary), especially investigating foods that contains phytochemicals known to have effects on cognitive functions. The influence of nutritional habits and/or the decision to not investigate these aspects should be included in the discussion.
  5. The Authors clarified what the “sanitary screening” consisted of and specified in the text their choice to include participants within a BMI range or 18.5 and 30 kg/m2; however, the “overweight” BMI range (25-30) might not be fully free of health complains. For this reason a screening carried out by a medical doctor is important to be fully confident in the “healthy” status of the participants.
  6. Considering that the product tested is commercially available, I wonder whether it has been tested for cognitive effects prior this study? Or its pharmaceutical properties are based on single specie human studies and the recent animal study?
  7. The Author included “Chronic Supplementation with a Mix of Salvia officinalis and Salvia lavandulaefolia Improves Morris Water Maze Learning in Normal Adult C57Bl/6J Mice” as reference for the information requested on the characteristics of the product. However, they still cannot provide information on the placebo composition, which I believe it is important to know.
  8. The authors answer the query and indicated in the test that no participant was supplemented less than 29 days. Perhaps it would be important to indicate that more than a few went over 29 days. Please discuss if and how this issue could have affected the results.
  9. I am still convinced that the age question, i.e. not considering age as a covariate in data analysis, is a big issue, when cognitive aspects are assessed. This problem must be discussed in the text, not in the footnote. This is a limitation of the study and should be acknowledged in the text and properly discussed.

RESULTS

  1. The authors rationalised the results. Could the authors report p value in table 2? It will be much easier to appreciate the information reported in the table without going back and forward in the text.
  2. It is still difficult to fully appreciate the amount of information provided in table and in graphs. Please check fig 10. It is still not clear whether the bottom graph is reporting an acute effect on day 29 or a chronic effect.
  3. It is appreciated that Authors reported that 69% of participants guessed correctly that they were in the “active condition”.  However, since this problem limits the achievement of the blinding condition, the Authors cannot limit their effort to just mention the problem, but they should discuss it and report this important limitation of the study.

DISCUSSION

Please discuss the points raised above and limitations of the study (see above)

SPECIFIC COMMENTS

  1. This point has not been addressed, so I riverberate my question: Fig 2: discrepancy with the procedure in the text regarding assessment of mood. What does VAS (the acronym in the figure 2 in the “cognitive demand battery” block) stand for?
  2. Line 426 Please delete “3.5.2. Corsi Blocks ‘Span Score’”
  3. Table 3, please define N/A

Author Response

To whom it may concern

Happy new year and thank you for the return of reviewer 2's round 2 comments. I hope you can appreciate the work which has been done to turn this around so quickly in the time-frame allowed in this 2nd round and that they have been completed to a satisfactory standard.

The only remaining issue seems to be the unknown of the exact inert placebo powder used but, as I say, my only contact at Nexira is now on paternity leave so I hope that this fact doesn't prevent the publication of this paper following such intensive revisions.

Thank you

Emma
